# Roles of Cytokines in the Temporal Changes of Microglial Membrane Currents and Neuronal Excitability and Synaptic Efficacy in ATP-Induced Cortical Injury Model

**DOI:** 10.3390/ijms22136853

**Published:** 2021-06-25

**Authors:** Bokyung Song, Sung-Joong Lee, Chong-Hyun Kim

**Affiliations:** 1Center for Neuroscience, Brain Science Institute, Korea Institute of Science and Technology, Seoul 02792, Korea; h13003@kist.re.kr; 2Neuroscience Program, Division of Bio-Medical Science and Technology, KIST School, Korea University of Science and Technology, Seoul 02792, Korea; 3Program in Neuroscience, Dental Research Institute, School of Dentistry, Seoul National University, Seoul 08826, Korea; sjlee87@snu.ac.kr

**Keywords:** ATP injury, cortex, microglia, cytokine, excitability, purinergic receptor, potassium channel

## Abstract

Cytokines are important neuroinflammatory modulators in neurodegenerative brain disorders including traumatic brain injury (TBI) and stroke. However, their temporal effects on the physiological properties of microglia and neurons during the recovery period have been unclear. Here, using an ATP-induced cortical injury model, we characterized selective effects of ATP injection compared to needle-control. In the damaged region, the fluorescent intensity of CX_3_CR1-GFP (+) cells, as well as the cell density, was increased and the maturation of newborn BrdU (+) cells continued until 28 day-post-injection (dpi) of ATP. The excitability and synaptic E/I balance of neurons and the inward and outward membrane currents of microglia were increased at 3 dpi, when expressions of tumor necrosis factor (TNF)-α/interleukin (IL)-1β and IL-10/IL-4 were also enhanced. These changes of both cells at 3 dpi were mostly decayed at 7 dpi and were suppressed by any of IL-10, IL-4, suramin (P2 receptor inhibitor) and 4-AP (K^+^ channel blocker). Acute ATP application alone induced only small effects from both naïve neurons and microglial cells in brain slice. However, TNF-α alone effectively increased the excitability of naïve neurons, which was blocked by suramin or 4-AP. TNF-α and IL-1β increased and decreased membrane currents of naïve microglia, respectively. Our results suggest that ATP and TNF-α dominantly induce the physiological activities of 3 dpi neurons and microglia, and IL-10 effectively suppresses such changes of both activated cells in K^+^ channel- and P2 receptor-dependent manner, while IL-4 suppresses neurons preferentially.

## 1. Introduction

Microglia are the innate immune cells in the brain which engage and respond to brain injury and infection by releasing various inflammatory cytokines, such as TNF-α, IL-1β, IL-6, IL-18, IL-12, interferon-γ, IL-4, IL-10, and transforming growth factor-β. They also respond to cytokines by their cognate receptors and affect neuronal functions during development and in many brain diseases [1,2,3,4,5,6,7,8,9,10]. Brain insults, such as trauma, ischemia and chronic neurodegenerative diseases, cause death of neurons and glial cells in the core damaged area. During the progression of brain injury, neuroinflammatory responses occur [11,12] where inflammatory mediators, such as cytokines and chemokines, are released by neurons and glial cells [12,13,14,15,16,17,18], meanwhile functions of survived neurons and glial cells are changed and new glial cells are generated [19,20,21]. Therefore, elucidating the mechanism of cytokine actions on neurons and microglia will help to facilitate the functional recovery process of damaged brain regions. Moreover, it will be important to find out whether any functional effects of cytokines on both cell types are constant or variable depending on the dose or the time course of the injury state, which reflects niche conditions.

Normally the levels of cytokines are low in CNS milieu, but they can be massively increased in response to brain injury. Pro-inflammatory cytokines such as TNF-α and IL-1β are known to contribute to cell death during the progression of neurodegenerative diseases [22,23,24,25]. Neuroprotective roles of anti-inflammatory cytokines, such as IL-4 and IL-10, have been shown in several brain disease conditions [26,27,28,29]. Another hallmark of brain tissue injury is the drastic release of ATP from damaged cells including neurons and from activated glial cells [30,31,32,33]. Though ATP has trophic effects on neurons and glial cells as an energy source and also acts as a neurotransmitter in a normal brain [32,34], extracellular ATP can be a danger signal, such as mitochondrial damage-associated molecular patterns (DAMPs or alamins), which recruit and activate microglia and induce acute neuronal cell death through the activation of purinergic receptors in brain injuries, such as traumatic brain injury and stroke [31,35,36,37], and seem to be involved in chronic inflammatory responses in age-related diseases, including neurodegenerative diseases, such as Alzheimer’s disease, Parkinson’s disease, etc., where the importance of the regulation of mitophagy and oxidative stress response has been increasingly revealed [38,39,40,41,42]. Thereby, blockers of purinergic receptors can improve recovery from some brain injuries [43,44]. 

Since the neuronal excitability and the physiological status of microglia implicate the functional status of both cells [45,46,47,48,49,50,51,52], increasing attention has been paid to the mechanism of cytokine actions on the physiological functions of both cells in healthy and pathological brain conditions. Studies on microglial membrane physiological properties have been tried in brain injury models, such as stab-wound injury [53], lipopolysaccharide-induced injury [54], ischemia [55,56], epilepsy [57], glioma [58] and Alzheimer’s disease [59]. However, most studies have focused on a specific time after the injury and few studies have looked over systematically the temporal effects of cytokine actions, as well as changes of microglial and neuronal physiological properties during the progression of brain injury models. Here, we have characterized the temporal changes of physiological features of both microglia and neurons in the damaged area for a month of post-injury period. In order to lessen the complexity due to factors involved in brain insult models, we adopted a simple ATP-induced cortical injury model, where acute death of neurons and glial cells occurs rapidly without inducing the secondary delayed neuronal death, which appears in brain injuries, such as TBI and stroke [60]. Using CX_3_CR1-GFP mice, we focused on the roles of ATP and cytokines in changes of neuronal and microglial physiological activities after ATP injection. We found that exogenous ATP was necessary for the microglia activation compared to needle-controls after the injury, and the sequential dominant actions of TNF-α and anti-inflammatory cytokines seemed to explain most temporal physiological changes observed in microglia and neurons, where the activities of both P2 receptors and K^+^ ion channels were critical.

## 2. Materials and Methods

### 2.1. Animals

For all experiments, 10−14-week-old male CX_3_CR1^GFP/+^ mice, generated on the C57BL/6 genetic background, were used (https://www.jax.org/strain/005582.html (accessed on 23 June 2021)). All animals were kept at room temperature under light/dark cycle and given ad-libitum feeding of food and water. Animal care and handling were performed in accordance to the guidelines set by the Animal Care and Use Committee of Korea Institute of Science and Technology (ACUCK).

### 2.2. Drugs (Chemicals)

All cytokines were purchased from PeproTech (Rocky Hill, NJ, USA, (IL-1β: #211-11B, TNF-α: #315-01A, IL-4: #214-14, IL-10: #210-10). These peptides were dissolved in water at 10 μg/mL and stored in aliquots at −80 °C. Bath cytokine concentrations were used at 10 ng/mL or 30 ng/mL. Blockers used were: CsOH solution (Sigma-Aldrich, St. Louis, MO, USA, #232041), quinine (1 mM, K^+^ channel blocker, Sigma-Aldrich, #Q1125), 4-aminopyridine (4-AP, 1 mM, Kdr channel blocker, Tocris, Bristol, UK, #0940), BaCl_2_ (200 µM, Kir channel blocker, Sigma-Aldrich, #217565), ML133 HCl (20 µM, Kir2.1 channel blocker, Tocris, #4549), Paxilline (50 µM, K_Ca_1.1 blocker, Tocris, #2006), A740003 (100 µM, P2X7R antagonist, Tocris, #3701) and Suramin hexasodium salt (500 µM, P2X&Y receptor antagonist, Tocris, #1472).

### 2.3. Stereotaxic Surgery for ATP Injection and BrdU Injection

After the mice were anesthetized with an intraperitoneal injection of Avertin (2,2,2-tribromethanol in 2-methylbutanol), the mice were placed on a stereotaxic frame (Stoelting Co., Wood Dale, IL, USA). 2 μL of ATP (50 mM) was delivered using an injection syringe (705RN 50 μL SYR, Hamilton Company, Reno, NV, USA) at a constant speed of 0.1 μL/min through a glass micro dispenser controlled by a microsyringe pump (Micro 4, World Precision Instruments (WPI), Sarasota, FL, USA). The injection site is, −1.8 mm anteroposterior (AP), ±1.5 mm medial lateral (ML), and −0.9 mm dorsal ventral (DV). A needle stab wound was used as a control. Bromodexoyuridine (BrdU, 300 mg/kg in saline, Sigma, St. Louis, MO, USA) was intraperitoneally injected once a day to quantify the proliferation and maturation of newborn cells. To see the maturation of newborn cells, BrdU was injected for 8 days, once per day, starting at 21 dpi.

### 2.4. Immunohistochemistry and Imaging

For the preparation of brain tissue, the mice were anesthetized with avertine (200 mg/kg, Sigma, St. Louis, MO, USA) and transcardially perfused with an ice-cold 0.1 M phosphate buffered saline solution (PBS, in mM; 137 NaCl, 2.7 KCl, 4.3 Na2HPO4, 1.47 KH2PO4; pH 7.4) and then 4% paraformaldehyde (PFA, Sigma, St. Louis, MO, USA). The brains were post-fixed overnight in 10% PFA at 4 °C, then incubated in 30% sucrose in 0.1 M PBS at 4 °C for 2 days. Following sucrose incubation, the brains were kept in cryoprotected tissue freezing in an OCT compound before being used for the experiment. A coronal section of 40 µm thickness was cut by using cryostat (HM525, Thermo Fisher Scientific, Waltham, MA, USA). Several tissues were used for free-floating immunohistochemistry and the remaining tissues were stored in a 0.1 M PBS/Glycine (1:1) solution at −20 °C for the next additional experiments.

For immunostaining, brain sections were washed with 0.1 M PBS three times. To block non-specific bindings, sections were incubated in 5% normal goat serum with 0.3% triton X-100 in PBS (Blocking solution) for 1 h at room temperature (RT) and then incubated in primary antibodies in a blocking solution for 16 h at 4 °C. After incubation, tissues were washed three times in 0.1 M PBS and incubated with fluorescent secondary antibodies in a blocking solution for 2 h at RT. After three times of the washing process, tissues were then stained with DAPI (1:1000) to stain cell nuclei for 20 min at RT and mounted on slides with a Dako fluorescent mounting medium. Antibodies used: IL-1b (AB_2124476), TNF-a (AB_302615), IL-4 (AB-629791), IL-10 (AB_10859554), Iba-1 (AB_839504).

### 2.5. Confocal Microscope Imaging and Analysis 

Fluorescent images were acquired using an Olympus confocal microscope (FV1000, Olympus, Japan) and ImageJ (NIH, Bethesda, MD, USA) software was used for measuring the soma area of the microglia and the fluorescent intensity of each cytokine and for counting the number of microglia labeled with DAPI or BrdU. To measure the cell body area of microglia, the images were acquired with 400× magnification and were opened with ImageJ (Appendix A). After the scale was set by “Analysis > Set scale”, the area of cell body was measured with the “polygon selection” function. To count cells labeled with DAPI or BrdU, each image of DAPI (405 nm) or BrdU (647 nm) bandwidth was generated separately and the “set threshold’ was used to convert them into the binary files. To measure and compare the fluorescent intensities of each cytokine expression in tissue samples from different days after injury (control, 3, 7, and 28 dpi), we used the same protocol for acquiring the fluorescence images from all samples. Since the cytokine expression of 3 dpi samples showed the strong signal intensity in general for all cytokines, the acquiring conditions for the fluorescence image of each cytokine in 3 dpi samples, where the laser power was set to minimize the saturation of each cytokine fluorescence, were used for all other samples from different days. To further reduce the sample variation, every image of all cytokines in the same tissue samples were taken on the same day under the same acquiring conditions. The core area was defined as the area where the GFP signal was significantly reduced at 3 dpi and the penumbra area was the surrounding area of the core region, where activated GFP (+) cells with bigger soma were visible. The damaged area includes both the core and the penumbra regions. The areas with 600× of confocal imaging data were analyzed by ImageJ.

### 2.6. Slice Preparation and Electrophysiology

The mice were anesthetized by halothane (2-bromo-2-chloro-1, 1, 1-trifluoroethane) and decapitated. Coronal 320-um-thick cortical brain slices were prepared by using a Vibratome (VT-1000S; Leica Biosystems, Buffalo Grove, IL, USA) and incubated for at least an hour in an oxygenated artificial cerebrospinal fluid containing (aCSF, in mM) 130 NaCl, 3.5 KCl, 1.25 NaH_2_PO_4_, 1.5 CaCl_2_, 1.5 MgCl_2_, 24 NaHCO_3_, and 10 D-glucose. Perforated whole-cell recordings were made on the soma of microglial cells or neurons in CX_3_CR1^GFP^ mice. Patch pipettes with a resistance of 6–7 MΩ were pulled using a pipette puller (Narishige, Tokyo, Japan, PC100) and filled with an internal solution (mM): 120 potassium gluconate, 15 potassium chloride, 5 sodium chloride, 1 magnesium ATP, 1 EGTA, 1 calcium chloride, and 10 HEPES; pH 7.2; 280–300 mOsm. For measurements of spontaneous excitatory and inhibitory postsynaptic currents (sEPSCs and sIPSCs), a Cs-gluconate internal solution was used (mM): 125 cesium gluconate, 8 potassium chloride, 10 HEPES, 1 EGTA, 4 MgATP and 0.5 NaGTP; pH 7.2~7.3; 280–295 mOsm. GFP (+) microglial cells were visualized with a 60X water objective on an IR-DIC microscope (Nikon Eclipse FN-1 microscope, Japan) and currents were recorded with an amplifier (MultiClamp 700B, Molecular Devices, San Jose, CA, USA), filtered at 2 kHz (low pass), digitized at 10 kHz (DigiData 1440A, Molecular Devices, San Jose, CA, USA), and collected and analyzed with pClamp10 (SCR_011323). For neurons, when the whole cell configuration was achieved at holding potential −65 mV, 500 ms current pulses were applied from −60 pA to +150 pA with an increment of 10 pA. Drugs were added after cytokines were pre-applied in a bath at least 30 min before patch recording. The rheobase was the minimum current step value injected to produce the first action potential from a neuron at the holding potential. At each current step of at +30 pA, +90 pA and +150 pA, the peak voltage amplitude was measured and the half-width was determined as the duration of the action potential at the voltage halfway between the threshold and the peak. Both sEPSCs and sIPSCs were recorded in normal ACSF with holding potential at −70 mV for EPSP and +10 mV for IPSP. For microglia, to investigate voltage-activated currents of microglia from a holding potential of −70 mV, a voltage ramp protocol was given from −50 mV to +150 mV for 1 s. Currents density was calculated by dividing membrane currents by membrane capacitance (pA/pF).

### 2.7. Statistical Analysis 

For statistical analysis, the GraphPad Prism 9 software (SCR_002798) was used. Results are given as means ± SEM, unless otherwise specified. A two-tailed *t*-test was performed and a two-way analysis of variance (ANOVA) were followed by post hoc Tukey’s multiple comparisons test when a significance at the level of 95% was found in the ANOVA test. * and # indicate the statistical significance of an experimental group to the control group and the 3 dpi group, respectively. *, # for *p* < 0.05, **, ## for *p* < 0.01, ***, ### for *p* < 0.001. The statistical results of Figure 1 and Figures 3–7 are given in Appendix A. The values of physiological parameters of Figures 5 and 6 are shown in Appendix A, respectively.

## 3. Results

### 3.1. Changes in CX_3_CR1-GFP (+) Cell Morphology and Number in Damaged Cortical Area 

After ATP injection into the motor cortex of CX_3_CR1^GFP/+^ mice, we observed the change of the size of the damaged area at 3, 7, and 28 dpi, during which the core region got smaller (Figure 1A,C). The soma size of the GFP (+) cells, mostly from the penumbra area, increased significantly at 3 dpi (Figure 1D). As defined, the GFP signal of microglia in the core region was significantly reduced at 3 dpi and then enhanced at 7 dpi (Figure 1E). The core cell density was increased at 7 dpi by ~36% and maintained (Figure 1F), implicating continuing proliferation/maturation of newborn cells or migration of cells. To test the proliferating activity at each dpi, BrdU was administrated once intraperitoneally 2 h before brain fixation. In control naïve cortical tissues, there were no BrdU (+) cells. BrdU (+) cells appeared even in the core at 3 dpi but disappeared at 28 dpi, implying no more new proliferating activity (Figure 1G,H). To estimate the contribution of newborn cells under maturation stages to the core cell population, BrdU was injected eight times starting at 21 dpi, though this would still give an underestimation. The result revealed much bigger BrdU (+) cell numbers even at 28 dpi, which could account for about half of the increased cell density in the core (Figure 1F,I). If no neurogenesis would occur within the core, the net increased core cell population would consist of newborn cells from microglia and non-microglial cells, and migrating monocytes (Figure 1J) (see Discussion). In case of needle-controls, there were much smaller damaged areas, less changes in the soma size of GFP (+) cells, and little effects on the single cell GFP intensity and the total cell density in the damaged region (Figure 1B–F).

### 3.2. Expression of Cytokines and Their Co-Labeling with Some Core CX_3_CR1 (+) Cells

Since cytokines were the markers of neuroinflammatory responses to brain injury, we looked at expressions of four cytokines, IL-1β/TNF-α and IL-4/IL-10, as representative pro- and anti-inflammatory cytokines (Figure 2A). These cytokines were rarely detected in naïve tissues. At the core, the expression of IL-1β and TNF-α seemed to peak at 3 dpi, while IL-4 and IL-10 did at 7 dpi, though they overlapped most periods (Figure 2B). The needle injection alone increased these cytokines with much lower and slower induction rates (Figure 2C), implicating a role of exogenous ATP in facilitating the expression of these inflammatory cytokines. Co-labeling of each cytokine with some core CX_3_CR1-GFP (+) cells show that these cytokines are expressed by microglia (Figure 2D).

### 3.3. Changes of Excitability and Synaptic Functions of Cortical Neurons after ATP Injection

After ATP injection, to examine the intrinsic excitability of survived neurons in the damaged area, action potential firing frequencies in response to depolarizing current steps were measured. The firing frequency of 3 dpi neurons was significantly increased while the excitability of needle-control neurons changed much less. (Figure 3A). The resting membrane potential (Vm) of neurons was not changed for one month (Figure 3B). The hyperpolarization-induced currents (Ih) were increased at 3 dpi (Figure 3C). The rheobase current was reduced at 3 dpi (Figure 3D). The membrane capacitance (Cm) was reduced in both ATP and needle-control groups but peaked at different times (Figure 3E). The membrane resistance (Rm) of neurons was not affected, except at 28 dpi needle-control (Figure 3F). The peak amplitude and half-width of action potentials seemed normal (Figure 3G,H). Interestingly, the kinetics of neuronal Cm change seemed correlated with those of pro-inflammatory cytokines.

To assess effects on the synaptic transmission efficacy, we measured sEPSCs and sIPSCs of neurons in damaged areas. We found reductions of both the frequency and the peak amplitude of sIPSC and an increase of the frequency of sEPSC of 3 dpi neurons (Figure 3I–M). The relative change in the frequency of sEPSC over sIPSC was increased at 3 dpi, which would promote the neuronal excitability (Figure 3N–O). The enhancement of 3 dpi neuronal firing frequency may be due to the combined effects, such as reductions of the rheobase and Cm and the enhanced spontaneous net excitatory synaptic drive. Collectively, these data show that 3 dpi neurons express a higher intrinsic firing status while receiving more frequent net excitatory synaptic inputs.

### 3.4. Changes of Microglial Membrane Currents in Response to Voltage Ramp Stimuli after ATP Injection

Since membrane conductance of microglia reflects functions of microglia, we measured the membrane current density of microglial cells in response to the voltage-ramp stimulus. Inward currents of 3 dpi microglia at −150 mV was increased approximately 4 times and outward currents at +50 mV were increased by more than 6 times over naïve cells, which then disappeared at 7 dpi (Figure 4A). Only a small increase of outward currents appeared in 3 dpi needle-controls. These results suggest roles of exogenous ATP in the enhancements of both inward and outward currents of 3 dpi microglial cells. The Vm of microglia was hyperpolarized at 3 dpi and then depolarized at 28 dpi (Figure 4B). The reversal potential (Vrev) and Rm were kept depolarized and reduced, respectively (Figure 4C,E). The Cm was increased only at 3 dpi (Figure 4F), consistent with the temporal increase of cell body size. There were no significant effects on these membrane properties of needle-control microglial cells. These data suggest that 3 dpi microglia show higher expressions of both inward and outward K^+^ ion channel currents, which also contribute to the reduction of membrane resistance, while having an increased membrane surface area, consistent with the soma size change (Figure 1D)

Next, to find the responsible ion channels, we measured the current responses in the presence of channel blockers. Enhancements of both inward and outward currents of 3 dpi microglia were suppressed effectively by diverse K^+^ channel blockers such, as Cs^2+^, quinine, 4-AP, Ba^2+^, ML133, Paxilline, and by P2 receptor blockers, such as A740003 and suramin (Figure 4F–N). The results suggest that P2 receptors on 3 dpi microglia are active and their activities are linked to the activation of related K^+^ channels.

### 3.5. Cytokines Modulate the Excitability of Naïve Neurons and Anti-Inflammatory Cytokines Suppress the Enhanced Excitability of 3 dpi Neurons

Since expressions of four cytokines were increased at 3 dpi cortical tissues, we checked the possibility whether those cytokines could affect the excitability of naïve cortical neurons (Figure 5A). A bath application of TNF-α increased significantly the firing frequency of naïve neurons, while reducing Cm only. ATP also increased the firing without effects on Cm (Appendix A). IL-1β had a marginal effect on the firing frequency with enhanced Ih (Figure 5B). IL-10 or IL-4 reduced the firing frequency, Rm, and Ih, while increasing the rheobase and Cm. The results suggest that the enhanced excitability of 3 dpi neurons can be reproduced by dominant net effects of pro-inflammatory cytokines, especially TNF-α, along with ATP on naïve neurons. Then, we checked whether anti-inflammatory cytokines were able to suppress the excitability of 3 dpi neurons. Either IL-10 or IL-4 effectively dropped the excitability of 3 dpi neurons back to the control level while reducing Ih (Figure 5C,D). These data suggest that ATP and TNF-α dominantly contribute to the enhancement of 3 dpi neuron excitability and IL-10 and IL-4 then dominantly induce the drop of excitability at 7 dpi and later.

### 3.6. Cytokines Affect Naïve Microglial Membrane Properties Differently and IL-10 Effectively Suppresses the Membrane Currents of 3 dpi Microglia

Microglia express cognate cytokine receptors [2]. At resting state, microglia experience a little concentration of extracellular cytokines; however, activated microglia are exposed to higher concentrations of cytokines (Figure 2). Therefore, we tested whether cytokines could activate naïve microglia and increase the membrane currents like 3 dpi microglia. While IL-1β at 10 ng/mL reduced both inward and outward membrane currents of naïve microglia and IL-4 also reduced the inward currents, other cytokines at 10 ng/mL or low concentrations of ATP (100–300 µM) had marginal effects on membrane currents of naïve microglia (Figure 6A). At 30 ng/mL, TNF-α and IL-1β increased and decreased both currents, respectively, meanwhile IL-10 increased the inward current but reduced the outward currents, though these changes were small compared to 3 dpi microglia. Interestingly, IL-4 had little effects on both currents of naïve microglia (Figure 6B). High concentrations of ATP (1 mM) caused a greater increase of inward current than the outward current. When we applied two (IL-1β + TNF-α or IL-4 + IL-10) or four cytokines (10 ng/mL each) together, co-application of IL-1β and TNF-α caused a small increase of both currents of naïve microglia (Figure 6C). The results suggest that IL-1 β or IL-4 alone at low concentrations has suppressive effects on the membrane currents of naïve microglia and high concentrations of both ATP and TNF-α are necessary for producing the current changes similar to those of 3 dpi microglia, which is consistent with the role of purinergic receptors in microglial activation [61].

Next, we asked whether anti-inflammatory cytokines could suppress the membrane currents of 3 dpi microglia (Figure 6F–J). IL-10 suppressed both inward and outward currents more effectively than IL-4. Interestingly, co-application of IL-10 with IL-4 inhibited the effects of IL-10 alone on the outward current and even enhanced the inward current. The results suggest that IL-10 alone is an effective suppressant of membrane currents of 3 dpi microglia. (See Appendix A for values of biophysical membrane properties at each condition).

### 3.7. Blockers of Purinergic Receptors or K^+^ Channels Suppress the Excitability of 3 dpi Neurons and TNF-α-Treated Naïve Neurons

Since neuronal P2Y receptor modulates activities of some K^+^ ion channels [62,63], we further asked whether blockers of P2 receptors or K^+^ ion channels could block the excitability of 3 dpi neurons (Figure 7A). Surprisingly, suramin or 4-AP completely suppressed the firing frequency of 3 dpi neurons to the control level, while converting Cm back to control levels (Figure 7E). Vm of 3 dpi neurons was hyperpolarized by these blockers, but the rheobase was not affected, suggesting an independence of the new rheobase from suramin- or 4-AP-sensitive ion channels (Figure 7C,D). Then we tested whether suramin or 4-AP could counteract the effect of TNF-α on naïve neurons. Either suramin or 4-AP blocked the TNF-α induced hyerexcitability. However, the mechanism of hyperexcitability of TNF-α-treated naïve neuron looked different from that of 3 dpi neurons. TNF-α reduced only Cm of naïve neurons, but both rheobase and Cm were reduced in 3 dpi neurons (Figure 7D,E). Moreover, co-application of TNFα with suramin to naïve neurons reduced only Rm, but the co-application with 4-AP affected Ih, Vm, and Cm. (Figure 7B–F). These results indicate the multiple signaling ways of modulating the action potential firing frequency of neurons and suggest that ATP-induced neuronal excitability changes involve activations of P2 receptors, K^+^ channels, and related cytokines.

In summary, using an ATP-induced cortical injury model, we showed the temporal kinetics of + TNF-α, IL-1β, IL-10, and IL-4 expressions, the proliferation and maturation of newborn cells, and changes of microglial membrane currents and the excitability and synaptic properties of neurons, where roles of P2 receptors, cytokines, and K^+^ ion channels are critically connected. The results suggest that TNF-α seems to play early dominant roles in activating both microglia and neurons at 3 dpi after ATP injection and anti-inflammatory cytokine IL-10 suppresses both activated cell types at 7 dpi and later, meanwhile IL-4 suppress neurons preferentially.

## 4. Discussion

In this study, we investigated the recovery of the neuronal and microglial physiological changes for a month’s period in ATP-induced injury models in a mouse cortex. We find a more selective understanding of the early ATP-induced contribution to the physiological changes of both neurons and microglia in the damaged brain area, which cannot be easily made by using conventional brain injury models for TBI and stroke.

### 4.1. ATP-Injection Injury Model Compared to Needle Stab-Wound Control

Compared to the current TBI and stroke animal models [64,65], the cortical injury by ATP injection is a simple and localized brain injury model [66,67]. The brain injury induced by ATP injection elicits combined effects from both mechanical (needle stab-wound) and chemical (ATP) insults. Here, we showed the clear difference between ATP-injection and needle-controls in changes of cytokine expression, cell population, microglial membrane currents and neuronal excitability in damaged areas. Since needle probe insertion into NAc caused ATP elevation only in the first 15 min [68], exogenous ATP must have contributed to initiate most effects observed in this study. After ATP injection, extracellular ATP levels at 3 dpi might be high enough to activate P2 receptors such that suramin could block the physiological changes of both microglia and neurons mostly.

### 4.2. Cell Population Changes in the Core Region of ATP-Injected Cortical Injury Site

We found that the damaged cortical area, including the core and penumbra regions. was continually reduced for a month after ATP injection. In the core region, the CX_3_CR1-GFP signal intensity was reduced at 3 dpi, probably due to the death of some GFP (+) cells, though the total cell number was not much changed and the soma size was even bigger. Both the GFP intensity and the cell number were increased at 7 dpi and maintained until 28 dpi, while the soma size was reduced back to control levels. Changes in the number of GFP (+) cells or GFP expression levels would contribute to the total GFP signal. Since CX_3_CR1 is a marker to microglia, monocytes, and macrophages [3,18], the migration or proliferation of these cells may change the cell number. Though ATP can induce the chemotactic migration of microglia [69,70], monocytes are the majority of macrophages in the core region of needle-control injury, which do not proliferate and their differentiation into microglia is disputed [19,71,72,73,74]. Moreover, resident microglia can proliferate locally in the brain [75,76]. After an 8-day-long BrdU injection, the core BrdU (+) cell density was approximately half of the increase in the core cell density at 28 dpi and only some core BrdU (+) cells were co-labeled with GFP (+) signals (Figure 1). Therefore, the remaining portion of the increased core cell population must consist of GFP (−) BrdU (+) cells like astrocytes/oligodendrocytes or GFP (+) BrdU (−) monocytes [73,77]. Additionally, our finding of the early newborn cells at 28 dpi reflects the maturation time course of adult newborn microglia [74].

### 4.3. Interactions among P2 Receptors, K^+^ Ion Channels, and Cytokines of 3 dpi Microglia

The interactions among subtypes of P2 receptors, cytokines, voltage-gated or Ca^2+^-activated K^+^ ion channels of microglia are involved in regulating membrane currents for cell functions, such as surveillance, migration, proliferation, and phagocytosis, and for releasing NO, ROS, cytokines, and chemokines under normal and neuroinflammatory conditions [45,49,50,51,52,78,79]. ATP activates microglia via P2Y and P2X7 receptors, which then release TNF-α and IL-1β via Ca^2+^ activity and increase K^+^ currents [61,80]. We found increases of both inward and outward K^+^ currents of 3 dpi microglia after ATP injection, which were sensitive to blockers of K^+^ ion channels, such as K_ir_2.1, K_Ca_1.1, and K_v_1.3, and purinergic receptors including P2X7R. Since the release of pro-inflammatory cytokines depends on the activation of K^+^ channels, suppression of activated microglial membrane currents might facilitate the recovery of damaged regions in ATP-induced brain injury [81]. The identification of the 4-AP-sensitive K^+^ ion channel and the suramin-sensitive purinergic receptor of microglia in this injury model remains for further study.

### 4.4. Cytokine Modulation of Neuronal Excitability and Synaptic Inputs

Effects of IL-1β and TNF-α on neuronal excitability involving Na^+^ and K^+^ channels and GABA_A_R currents were complex, depending on cell type, dose, and exposure time [82,83,84]. We found that 3 dpi neurons had an enhanced firing frequency and Ih, while having the reduced rheobase and Cm. On the other hand, TNF-α increased the firing frequency of naïve neurons while decreasing only Cm a bit. These data suggest changes in 3 dpi neurons are not induced by TNF-α alone. Interestingly, IL-4 or IL-10 reduced the firing frequency of both naïve and 3 dpi neurons via increasing the rheobase and Cm in common, while other properties were affected differently (Appendix A). Moreover, 4-AP or suramin suppressed the firing of TNF-α-treated neurons while reducing Rm in common only. Likewise, both blockers suppressed the firing of 3 dpi neurons while affecting Vm and Cm in common but Rm differently (Figure 7). These results clearly reflect multiple ways of modulating the firing frequency of neurons through interactions among cytokines, P2 receptors, and K^+^ channels, in a cell-state or niche dependent manner. The mechanism of 4-AP inhibition of the firing of TNF-α-treated neurons remains for future study. Furthermore, we found a relatively enhanced spontaneous excitatory synaptic drive to 3 dpi neurons. The increase of the sEPSC frequency of 3 dpi neurons might be explained by dominant TNF-α and IL-10 actions on the surface AMAP receptor trafficking against the opposing actions of IL-1β [82,84,85,86]. The decrease of the sIPSC amplitude and frequency of 3 dpi neurons might be due to dominant TNF-α, IL-1β and IL-10 actions on GABAergic currents of neurons over IL-4 [82,85,87,88].

### 4.5. Roles of TNF-α, IL-1β, IL-4, and IL-10 in ATP-Induced Cortical Brain Injury

The final effects of TNF-α and IL-1β can be either neurotoxic or neuroprotective depending on concentration and disease state [23,82,83,84]. We saw dose-dependent changes of single cytokine effects on microglial membrane currents (Figure 6). Interestingly, we found low concentrations of IL-1 β alone inhibited naïve microglial membrane currents, which would suppress the activation of microglia. Therefore, IL-1β acts like a suppressing cytokine to both microglial membrane currents and neuronal AMPA receptor currents in this model. Though expressions of pro- and anti-inflammatory cytokines are mostly overlapped in damaged cortical regions after ATP-injection, their subtle temporal kinetic difference suggests that early effects are produced dominantly by pro-inflammatory cytokines and later effects are produced by anti-inflammatory cytokines. Among cytokines tested, we found more select roles of TNF-α and IL-10 in modulating the excitability of cortical neurons, sensitively but oppositely. In addition, naïve microglia seem more sensitive to a low concentration of IL-4 and insensitive to IL-10, while activated microglia are more sensitive to IL-10 than to IL-4. Since IL-10 inhibits the release of pro-inflammatory cytokines and impedes TNF-α mediated inflammatory changes of microglia and neurotoxicity in injury models [27,89], these results together suggest IL-10 as an effective suppressant for the microglial activation, which thereby would ameliorate the progression of the deteriorating neuroinflammatory responses after the injury.

## 5. Conclusions

Most neurodegenerative brain disorders elicit neuroinflammatory responses during their pathological progression. The roles of microglia upon neuronal functions after inflammatory brain insults have been unclear partly due to complexities of cytokines and their regulations in brain disease models. Understanding the role of cytokines on neurons and microglia will be important for facilitating the functional recovery of damaged brain regions. Here, using a simple ATP-induced cortical injury model, a systematic characterization of the changes of membrane currents of microglia and neuronal excitability has been made for a period of a month. We find that ATP and TNF-α are necessary for the activation of microglia and the neuronal hyperexcitability at 3 dpi. These effects are clearly selective compared to those of needle-controls. The results also suggest IL-10 as an effective suppressant for the physiological changes of both activated cell types at 3 dpi, where activities of 4-AP-sensitive K^+^ ion channels and suramin-sensitive P2 receptors are critical. Though limited in the scope, this study on the ATP-mediated physiological changes of neurons and microglial cells will help understand the more complex mechanism of the progression of pathological brain injuries like TBI, stroke, and chronic neurodegenerative diseases such as Alzheimer’s disease and Parkinson’s disease.

## Figures and Tables

**Figure 1 ijms-22-06853-f001:**
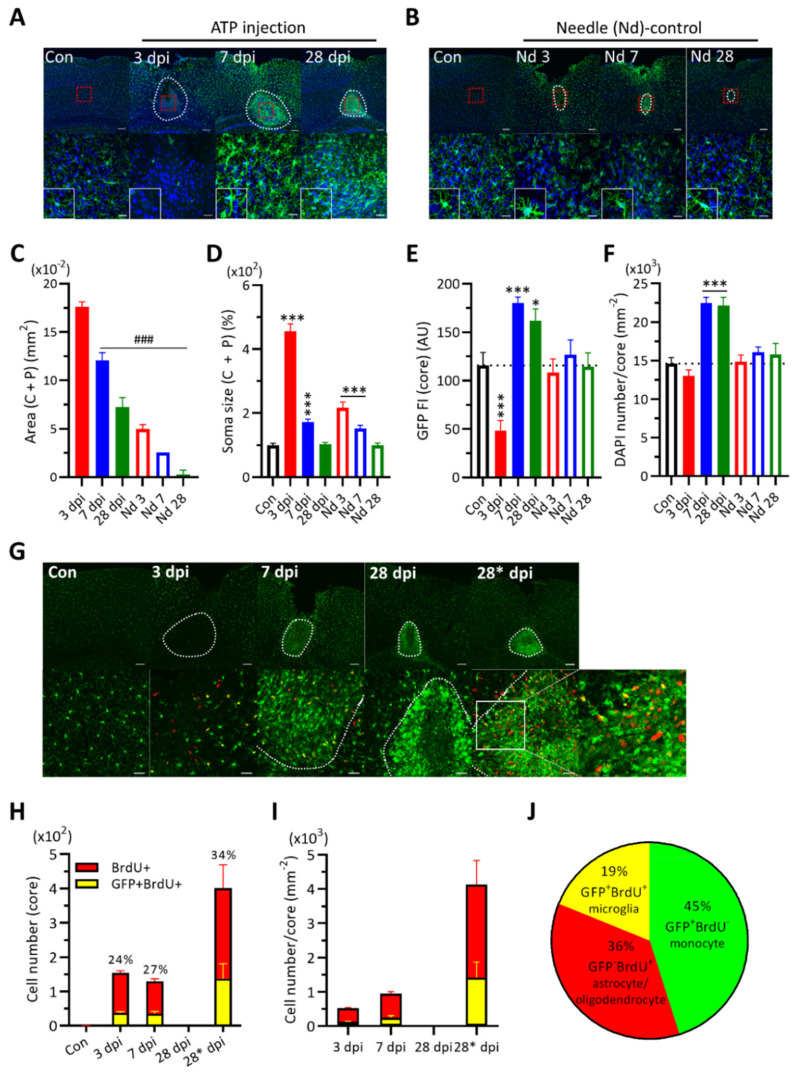
ATP injection causes significant cellular changes compared to needle-control in cortical area. Two-tailed *t*-test for comparing the experimental group to the control group (*) or to the 3 dpi group (#) (*, # for *p* < 0.05; **, ## for *p* < 0.01; ***, ### for *p* < 0.001). (**A**). *Upper*: Representative images of CX_3_CR1-GFP expression in cortical tissues from ATP injected mice (Control, 3rd, 7th, 28th dpi). DAPI in blue. White dotted line indicates the damaged area (core + penumbra); red square indicates the area for high magnification image below. *Lower*: high magnification images. *Scale bars*, 100 μm (*upper*) and 20 μm (*lower*). *Inset*, representative image of CX_3_CR1-GFP (+) cells in the core. (**B**). Needle (Nd)-control cortical tissue data (Control, Nd 3rd, Nd 7th, Nd 28th dpi). Same as in (**A**). (**C**). Measurement of damaged areas by ImageJ. Two-tailed *t*-test. Open bars indicate naïve or needle-control tissue data and filled bars indicate ATP-injected tissue data (**C**–**F**). (**D**). Soma size of CX_3_CR1-GFP (+) cells in damaged areas. Two-tailed *t*-test. (**E**). GFP fluorescence intensity measurement per core area. Two-tailed *t*-test. (**F**). DAPI (+) cell density in the core. Two-tailed *t*-test. (**G**). *Upper*: Representative images showing BrdU (+) cells (red) in the damaged area after ATP injection. Con, 3 dpi, 7 dpi, 28 dpi for BrdU injection once; 28* dpi for BrdU 8× injection. White dotted line indicates the core area for analysis. *Lower*: Magnified images. *Scale bars*, 100 μm (upper) and 20 μm (lower). *Far right* is the expanded image of the square region (white line). (**H**). The number of BrdU (+) cells in the core and the portion (%) of GFP (+) BrdU (+) cells among BrdU (+) cells. (**I**). The densities of BrdU (+) cells (in red) and BrdU (+) GFP (+) cells (in yellow) in the core. (**J**). The pie graph of the estimated portion of population of each cell types in the core area at 28* dpi.

**Figure 2 ijms-22-06853-f002:**
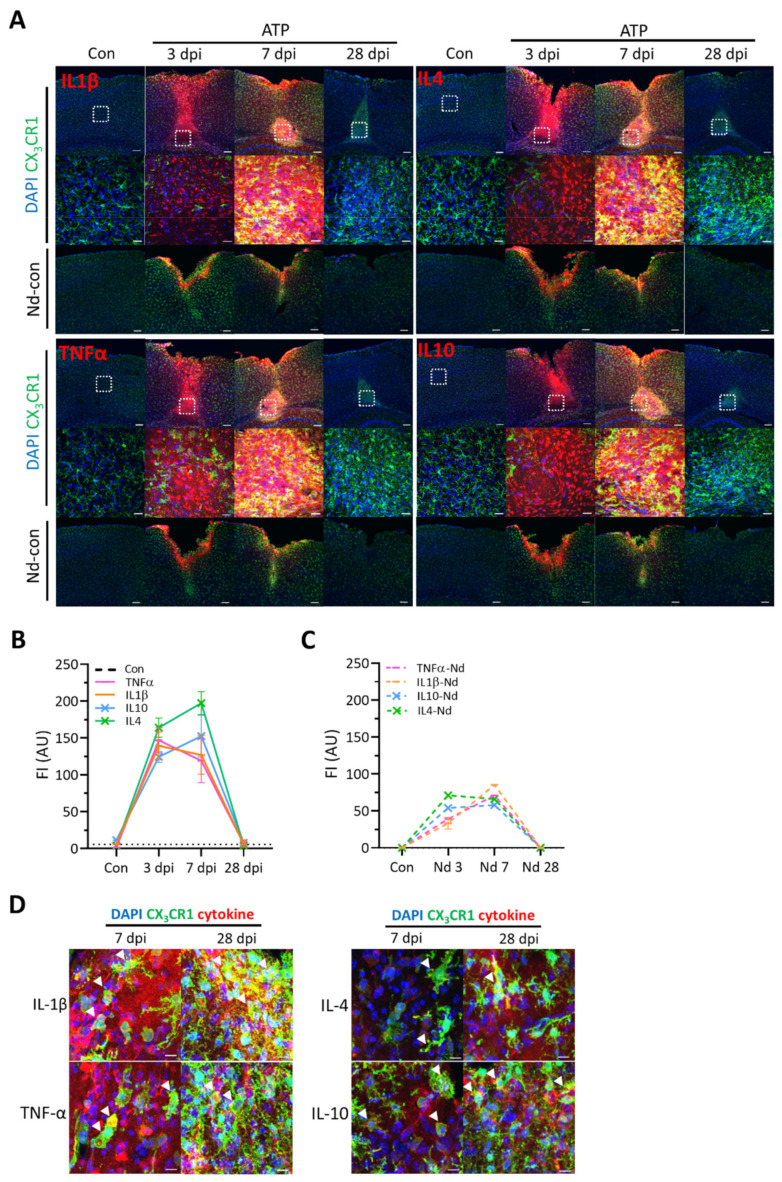
Cytokine expressions after ATP injection and their co-labeling with CX_3_CR1-GFP (+) cells. (**A**). *Top*: Representative images of immunohistochemistry of cytokines, IL-1β, TNF-α, IL-4, and IL10, after ATP injection. *Middle*: Magnification of white square area in *Top*. DAPI (blue), cytokine (red), and CX_3_CR1 (green). *Scale bars*, 100 µm (*Top*) and 20 µm (*middle*). *Bottom*: Needle-control images of each cytokines. (**B**). Quantification of the fluorescence intensity of each cytokine at days after ATP injection. Con (black dash line) indicates the intensity of each cytokines in control tissues. (**C**). Quantification of fluorescence intensity of each cytokines at days after needle stab wound. (**D**). Representative images of the co-labeling (white triangles) of each cytokine with activated CX3CR1-GFP (+) microglia in the core of 7 dpi and 28 dpi tissues. *Scale bar*, 10 µm.

**Figure 3 ijms-22-06853-f003:**
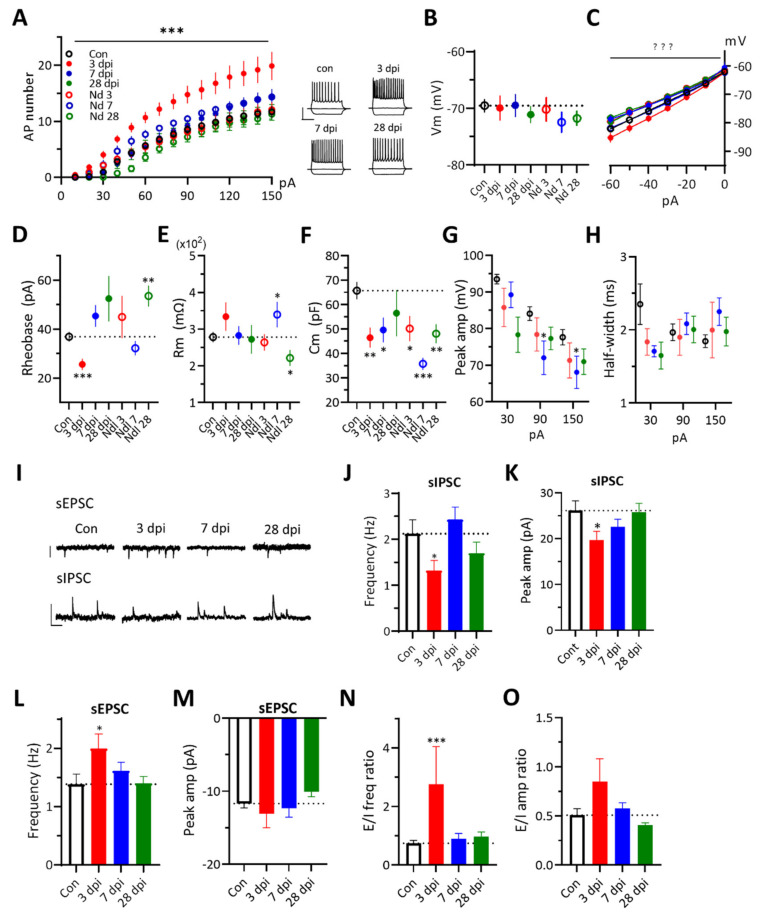
Changes of neuronal excitability and synaptic transmission properties for a month period after ATP injection. Two-way ANOVA with Tukey’s test or two-tailed *t*-test for comparing the experimental group to the control group (*, *p* < 0.05; **, *p* < 0.01; ***, *p* < 0.001). (**A**). The number of action potential firings in neurons at the indicated dpi to 500 msec depolarization pulses compared with needle-controls. Two-way ANOVA with Tukey’s test. *Insets*, Example traces of cortical neuron firing to −60, 0, +120 pA current injections. The same color and symbol of each condition are used from A to H below. (**B**). Resting membrane potential measurements. Two-tailed *t*-test. (**C**). I–V plot of neurons to hyperpolarizing current steps. Two-way ANOVA with Tukey’s test. (**D**). Rheobase measurements. Two-tailed *t*-test. (**E**). Membrane resistance measurements. Two-tailed *t*-test. (**F**). Membrane capacitance measurements. Two-tailed *t*-test. (**G**,**H**). Measurements of the peak amplitude and the half-width of 1st action potential spike generated at +30, +90 and +150 pA injections. Two-tailed *t*-test. (**I**). Example traces of sEPSCs recorded at −70 mV and sIPSC recorded at +10 mV of holding potentials in control and ATP injected groups. *Scale bars*, 20 pA (sEPSC), 50 pA (sIPSC), 200 ms. (**J**,**K**). The frequency and peak amplitude of sIPSCs. Two-tailed *t*-test. (**L**,**M**). The frequency and the peak amplitude of sEPSCs. Two-tailed *t*-test. (**N**). The frequency ratio of sEPSC over sIPSC. Two-tailed *t*-test. (**O**). The amplitude ratio of sEPSC over sIPSC. Two-tailed *t*-test.

**Figure 4 ijms-22-06853-f004:**
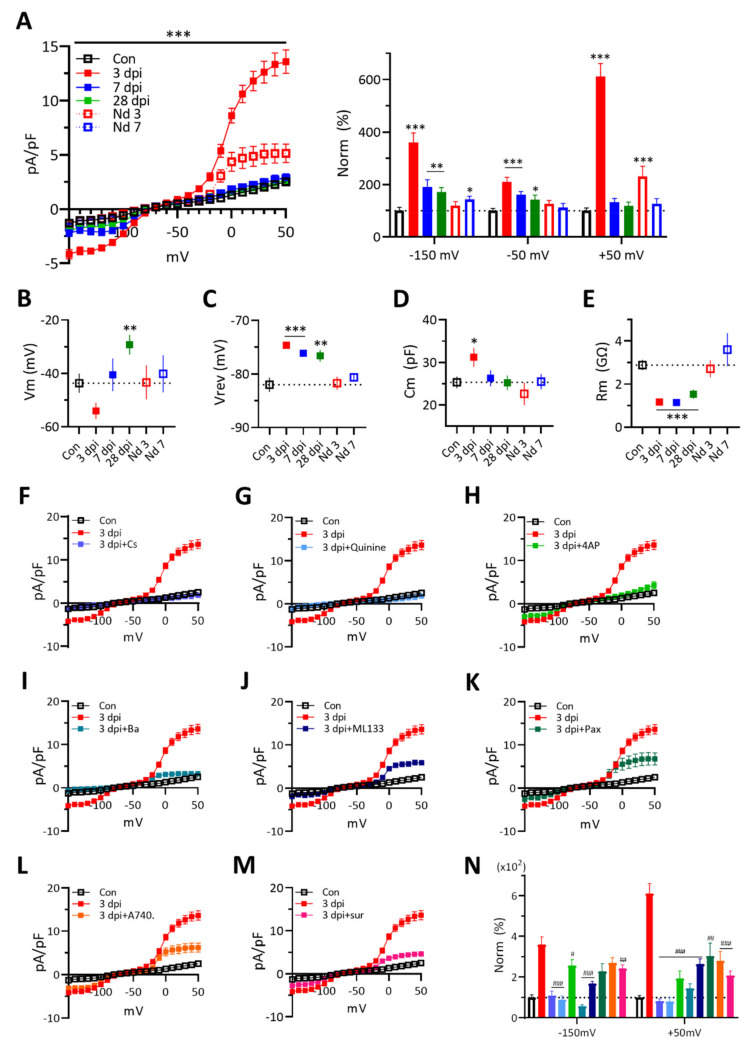
Changes of membrane current density of GFP (+) microglia in response to a voltage-ramp after ATP injection. Two-way ANOVA with Tukey’s test or two-tailed *t*-test for comparing the experimental group to the control group (*) or to the 3 dpi group (#) (*, # for *p* < 0.05; **, ## for *p* < 0.01; ***, ### for *p* < 0.001). (**A**). *Left*: Voltage-activated current density of microglial cells at different conditions. The current density was averaged by 10 mV increment. Two-way ANOVA with Tukey’s test. *Right*: Normalized current densities at −150, −50 and +50 mV. Two-tailed *t*-test. (**B**). Resting membrane potential of microglia. Two-tailed *t*-test. (**C**). Reversal potential of microglia. Two-tailed *t*-test. (**D**). Membrane capacitance of microglia. Two-tailed *t*-test. (**E**). Membrane resistance of microglia. Two-tailed *t*-test. (**F**–**M**). Membrane current density changes of 3 dpi microglia to internal cesium (Cs^2+^) solution (F) or in bath containing quinine (1 mM) (**G**), 4-AP (1 mM) (**H**), BaCl_2_ (200 µM) (**I**), ML133 (20 µM) (**J**), paxilline (50 µM) (**K**), A740003 (100 µM) (**L**), suramin (500 µM) (**M**,**N**). Normalized current densities of microglia from (**F**–**M**) at −150, −50 and +50 mV. The bar color indicates the same color of each molecule tested in (**F**–**M**). Two-tailed *t*-test.

**Figure 5 ijms-22-06853-f005:**
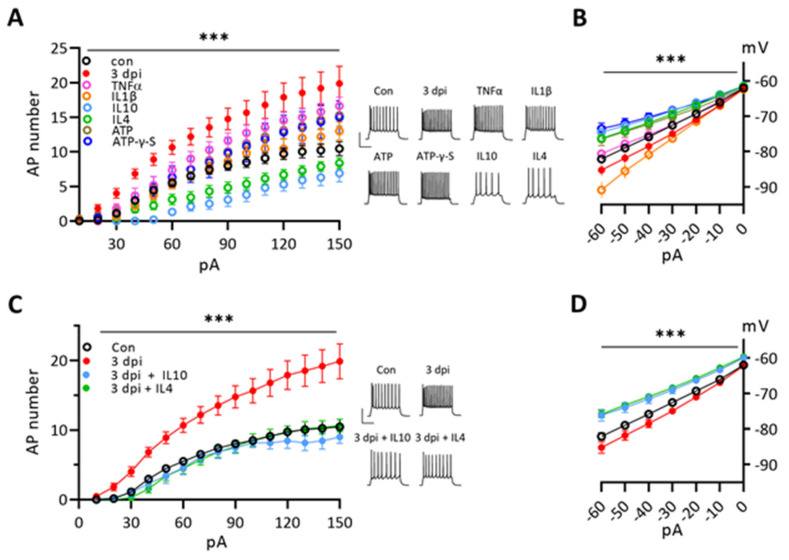
Effects of cytokine and ATP on the excitabilities of naïve neurons and 3 dpi neurons. Two-way ANOVA with Tukey’s test for comparing the experimental group to the control group (***, *p* < 0.001). (**A**). Effects of each cytokine and ATP on the number of action potential firings of naïve neurons to 500 msec depolarization pulses compared with those of naïve controls and 3 dpi neurons. Each cytokine (10 ng/mL), ATP (1 mM), ATP-γ-S (100 µM). Two-way ANOVA with Tukey’s test. *Insets*, Example traces of cortical neuron firing to +120 pA current injection under each conditions. *Scale bars*, 50 mV and 200 ms. The same color and symbol of each condition are used in (**A**,**B**). (**B**). Effects of each cytokine and ATP on I–V plot of naïve neurons to hyperpolarizing current steps compared with naïve controls and 3 dpi neurons. Two-way ANOVA with Tukey’s test. (**C**). Effects of IL-10 and IL-4 on the action potential firing numbers of 3 dpi neurons compared to naïve controls. Two-way ANOVA with Tukey’s test. *Insets and scale bars*, the same conditions in (**A**). The same color and symbol of each condition are used in (**C**,**D**). (**D**). Effects of IL-10 and IL-4 on I–V plot of 3 dpi neurons to hyperpolarizing current steps compared with naïve controls. Two-way ANOVA with Tukey’s test.

**Figure 6 ijms-22-06853-f006:**
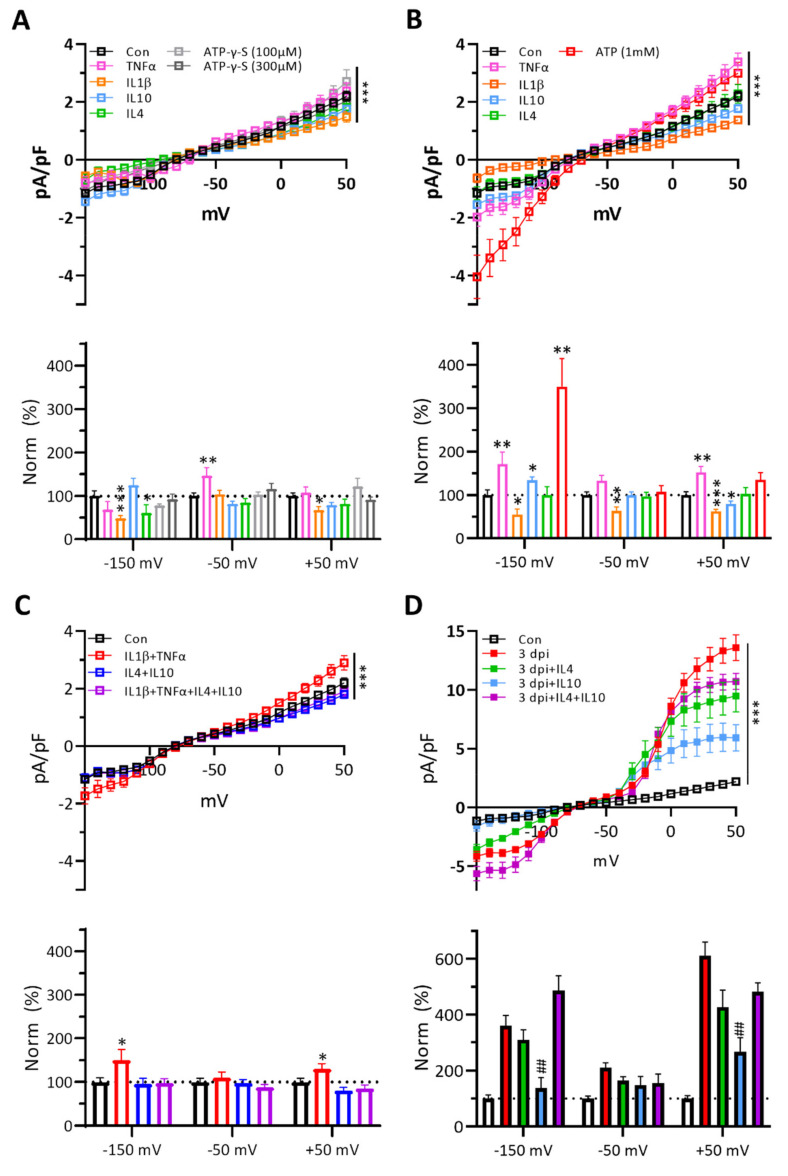
Effect of each cytokine and ATP on the membrane current density of GFP (+) naïve and 3 dpi microglia in response to a voltage-ramp. The current density was averaged by 10 mV increment. Two-way ANOVA with Tukey’s test or two-tailed *t*-test for comparing the experimental group (*) or to the 3 dpi group (#) (*, # for *p* < 0.05; **, ## for *p* < 0.01; ***, ### for *p* < 0.001). Two-way ANOVA and Tukey’s test, *** *p* < 0.001. (**A**). *Top*: Effects of each cytokine at a low-concentration (10 ng/mL) and ATP-γ-S (100 and 300 µM) on the voltage-activated current density of GFP (+) naïve microglial cells. Two-way ANOVA with Tukey’s test. *Bottom*: Normalized current densities at −150, −50 and +50 mV. Two-tailed *t*-test. (**B**). *Top*: Effects of each cytokine at a high-concentration (30 ng/mL) and ATP (1 mM) on the voltage-activated current density of GFP (+) naïve microglial cells. Two-way ANOVA with Tukey’s test. *Bottom*: Normalized current densities at −150, −50 and +50 mV. Two-tailed *t*-test. (**C**). *Top*: Combined effects of two or four cytokines (each 10 ng/mL) on the voltage-activated current density of GFP (+) naïve microglial cells. Two-way ANOVA with Tukey’s test. *Bottom*: Normalized current densities at −150, −50 and +50 mV. Two-tailed *t*-test. (**D**). *Top*: Effects of IL-4, IL-10 or both, at low-concentrations (10 ng/mL) on the voltage-activated current density of GFP (+) 3 dpi microglial cells. Two-way ANOVA with Tukey’s test. *Bottom*: Normalized current densities at −150, −50 and +50 mV. Two-tailed *t*-test.

**Figure 7 ijms-22-06853-f007:**
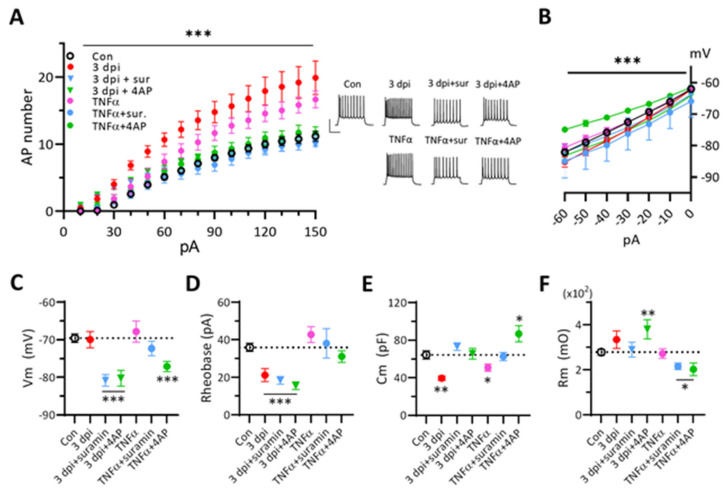
Effects of P2 receptor blocker or K^+^ channel blocker on TNF-α induced excitability changes of naïve neurons compared to ATP-injection induced excitability changes of 3 dpi neurons. Two-way ANOVA with Tukey’s test or two-tailed *t*-test for comparing the experimental group to the control group (*, *p* < 0.05; **, *p* < 0.001; ***, *p* < 0.0001). (**A**). Effects of suramin or 4-AP on the number of action potential firings of TNF-α-treated naïve neurons or 3 dpi neurons in response to 500 msec depolarization pulses compared with naïve controls. TNF-α (10 ng/mL), suramin (500 µM), 4-AP (1 mM). Two-way ANOVA with Tukey’s test. *Insets*, Example traces of cortical neuron firing to +120 pA current injection under each conditions. *Scale bars*, 50 mV and 200 ms. The same color and symbol of each condition are used in (**A**–**F**). (**B**). Effects of suramin or 4-AP on I–V plot to hyperpolarizing current steps of TNF-α-treated naïve neurons or 3 dpi neurons. Two-way ANOVA with Tukey’s test. (**C**). Resting membrane potential measurements. Two-tailed *t*-test. (**D**). Rheobase measurements. Two-tailed *t*-test. (**E**). Membrane resistance measurements. Two-tailed *t*-test. (**F**). Membrane capacitance measurements. Two-tailed *t*-test.

## Data Availability

The data that support the findings of this study are available from the corresponding author upon reasonable request.

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
