# Peer review of "Roles of Cytokines in the Temporal Changes of Microglial Membrane Currents and Neuronal Excitability and Synaptic Efficacy in ATP-Induced Cortical Injury Model"

_ijms, 2021, doi:10.3390/ijms22136853_

Round 1
Reviewer 1 Report
%MCEPASTEBIN%
Overall, this study is well designed and well-written, but for some examples of unclear sentences and editorial errors. My main critique of this study is the lack of information on the relevance of the ATP-induced injury model. The study would benefit from this information with proper citations to support the validity and relevance of this model in the introduction and some discussion around what the implications of the authors’ findings are for conditions characterized by ATP-induced injury in the discussion section. Other more specific comments, it is unclear in the methods and results sections what the t-tests are comparing. Presumably they are comparing the experimental groups to controls but this needs to be explicitly stated in the methods and in the figure legends to enhance the readability of the paper. In addition, in places where the authors performed multiple t-tests, a multiple comparisons correction needs to be performed (e.g., data in Figure 1). The authors need to do that provide the adjusted p-values. There are some issues with text clarity as well as spelling and editorial errors that need to be addressed before publication. Proof-reading the manuscript should remedy some of these issues. For example, Line 19- the sentence starting with “Such changes…” is awkward and needs to be rewritten for clarity. Line 21-“ naïve both cells” – what do the authors means? Do they mean naïve neurons and microglia? Line 47- respond should be response Line 130- were should be was Line 361- improper use of the word “ironically”. I think the authors mean interestingly or something to that effect. There are other examples of these kinds of error that should be addressed to enhance the clarity of the paper.
Author Response
we thank Reviewer 1 for the constructive comments on the manuscript. Below are the replies (in red) we provide to each of questions and comments from the reviewer 1.
Reviewer #1:
Overall, this study is well designed and well-written, but for some examples of unclear sentences and editorial errors.
My main critique of this study is the lack of information on the relevance of the ATP-induced injury model. The study would benefit from this information with proper citations to support the validity and relevance of this model in the introduction and some discussion around what the implications of the authors’ findings are for conditions characterized by ATP-induced injury in the discussion section.
Reply) As suggested, we add more information and reference regarding on the relevance of the ATP-injury model in the introduction and the possible benefit of this study in the discussion.
-“Another hallmark of brain injury… and induces inflammatory responses [31, 34-37]” in the introduction is now rewritten to “Another hallmark of brain injury… where the importance of the regulation of mitophagy and oxidative stress has been increased [38-42]”.
-“…where only acute death of neurons and glial cells… like TBI and stroke [60].” is added in the introduction.
-“In this study, we investigated… models for TBI and stroke.” is now added in the Discussion.
-“Though limited in the scope, this… ATP-induced cortical injury model.” is now rewritten to “Though limited in the scope, this… Alzheimer’s disease and Parkinson’s disease.” in the conclusions.
-References below are now added in the revised manuscript:
Zheng et al. (1991), Jeong et al. (2013a), Fivenson et al. (2017), Grizioli and Pugin (2018), Singh et al. (2019), Cai and Jeong (2020).
Other more specific comments, it is unclear in the methods and results sections what the t-tests are comparing. Presumably they are comparing the experimental groups to controls but this needs to be explicitly stated in the methods and in the figure legends to enhance the readability of the paper.
Reply) As suggested, the experimental group data were compared either to the control group data (*) or to the 3 dpi data (#). This is now clearly restated in the statistical analysis section of Methods and in each figure legend.
- “* and # indicate…” is restated in the statistical analysis section of Methods.
In addition, in places where the authors performed multiple t-tests, a multiple comparisons correction needs to be performed (e.g., data in Figure 1). The authors need to do that provide the adjusted p-values.
Reply) We have not done multiple t-tests to the data in Figure 1. Only two-tailed t-test was done. This is now clearly restated in the Figure legend.
There are some issues with text clarity as well as spelling and editorial errors that need to be addressed before publication. Proof-reading the manuscript should remedy some of these issues. For example,
Reply) Grammar and spelling were checked and changes were made.
Line 19- the sentence starting with “Such changes…” is awkward and needs to be rewritten for clarity.
Reply) “Such changes …” is now restated as “These changes…”.
Line 21-“ naïve both cells” – what do the authors means? Do they mean naïve neurons and microglia?
Reply) “naïve both cells” is now restated as “both naïve neurons and microglia”.
Line 47- respond should be response
Reply) “respond” is now changed to “response”.
Line 130- were should be was
Reply) “were” is now changed to “was”.
Line 361- improper use of the word “ironically”. I think the authors mean interestingly or something to that effect. There are other examples of these kinds of error that should be addressed to enhance the clarity of the paper.
Reply) “Ironically” is now changed to “Interestingly”.
-and more...
Below are the things to be done by Editorial office.
-Figure 3 has been updated due to a labeling error in C.
-Supplementary Table 1 has been edited by adding the type of statistics into the column of Figure 7.
Reviewer 2 Report
1. This manuscript still needs a proofreading in English grammar and some spelling.
2. The format of the manuscript can be improved for better understanding. The Discussion section is too redundant and some statements were repetitions of the Results. The author should merge Results and Discussion and brief the statements.
3. Some statements are not appropriate scientific language. The author should use a plain scientific language in the text. Some paragraphs are too short and should be merged with neighboring paragraphs.
4. Cytokine upregulation in the model is a key finding in this research. However, immune staining of the cytokine in slice may only reflect the expression of cytokine precursors. To confirm this improtant finding, the author should supply qRT-PCR and Elisa data.
Round 2
Reviewer 2 Report
The manuscript had been improved and can be accepted for publication.